# Weight Gain and Increased Body Mass Index in Patients with Hepatitis C after Eradication Using Direct-Acting Antiviral Therapy in Taiwan

**DOI:** 10.3390/diagnostics14020213

**Published:** 2024-01-19

**Authors:** Chun-Hsien Chen, Yung-Yu Hsieh, Wei-Ming Chen, Chien-Heng Shen, Kuo-Liang Wei, Kao-Chi Chang, Yuan-Jie Ding, Sheng-Nan Lu, Chao-Hung Hung, Te-Sheng Chang

**Affiliations:** 1Department of Gastroenterology and Hepatology, Division of Internal Medicine, Chang Gung Memorial Hospital, Chiayi 61363, Taiwan; richardc27629@gmail.com (C.-H.C.);; 2College of Medicine, Chang Gung University, Taoyuan 33303, Taiwan; 3Department of Gastroenterology and Hepatology, Division of Internal Medicine, Kaohsiung Chang Gung Memorial Hospital, Kaohsiung 83301, Taiwan

**Keywords:** hepatitis C virus (HCV), direct-acting antiviral therapy, weight gain, obesity

## Abstract

Few studies have reported weight gain in patients with hepatitis C virus (HCV) infection treated with direct-acting antiviral agents (DAAs). This retrospective cohort study identified factors associated with substantial weight gain after DAA treatment in Taiwan. This study involved patients treated using DAAs at the Chiayi and Yunlin branches of Chang Gung Memorial Hospital from 1 January 2017 to 31 October 2020. Body weight data were collected at the start of DAA therapy and 2 years after the confirmation of a sustained virologic response. We performed multiple logistic regression to evaluate the clinical and laboratory parameters associated with a large body mass index (BMI) increase (≥5%). The mean BMI was 25.56 ± 4.07 kg/m^2^ at baseline and 25.77 ± 4.29 kg/m^2^ at the endpoint (*p* = 0.005). A considerable reduction in fibrosis-4 (FIB-4) score was a significant predictor of a large BMI increase (OR: 1.168; 95% CI: 1.047–1.304, *p* = 0.006). By contrast, older age (OR: 0.979; 95% CI: 0.963–0.996, *p* = 0.013) and a higher baseline BMI (OR: 0.907; 95% CI: 0.863–0.954, *p* < 0.001) were associated with a reduced risk of a large increase in BMI at the endpoint. In summary, a larger BMI increase was closely associated with a younger age, lower baseline BMI, and higher FIB-4 score reduction. Notably, differences in DAA regimens did not affect outcomes. Future studies are needed to elucidate the long-term effects and metabolic outcomes associated with this body weight change and investigate the exact underlying mechanisms.

## 1. Introduction

Chronic hepatitis C virus (HCV) infection can cause various liver diseases, including cirrhosis, liver failure, and liver cancer. Furthermore, it can interfere with host metabolism, increasing the risk of diabetes, chronic kidney disease, cardiovascular disease, and stroke. However, successful treatment of HCV infection reduces the risk of intrahepatic and extrahepatic disease [1,2]. Unlike treatment involving interferon and ribavirin, direct-acting antiviral (DAA) therapy directly interrupts viral replication, has fewer side effects, has a shorter treatment course, and has an ultra-high cure rate (>98%) [3]. In the future, the discussion about HCV will shift from how to treat HCV successfully to how to appropriately track and monitor changes in the liver and related metabolic diseases after successful treatment.

Hepatic steatosis is a characteristic histological manifestation of HCV infection. HCV induces excess production of lipid droplets in infected hepatocytes and uses these droplets as scaffolds for virus assembly, leading to hepatic steatosis [4]. Hepatic steatosis is associated with insulin resistance, liver fibrosis, oxidative stress, and an increased incidence of hepatocellular carcinoma (HCC) [5,6,7]. Notably, insulin resistance is a hallmark feature of obesity and type 2 diabetes [8]. The successful eradication of HCV should lead to these problems being reversed. However, some studies have reported that patients may gain weight after HCV eradication [9,10,11,12]. In addition, there is no relevant literature studying this issue in Taiwan. Therefore, we conducted this study to elucidate the relationship between HCV eradication by DAA and post-treatment body weight changes, identify possible predictive factors, and improve our long-term care of patients after successful HCV treatment.

## 2. Materials and Methods

### 2.1. Study Design and Patient Population

In Taiwan, patients with HCV infection receive DAA treatment through a nationwide government-funded program launched in 2017. All patients with HCV and proven active viremia, regardless of the duration and severity of liver disease, are eligible for DAA treatment, with the exception of patients with advanced- or terminal-stage disease or with a limited life expectancy of <6 months. As the availability of different DAAs evolved over time, the approved DAAs in Taiwan include daclatasvir/asunaprevir with or without ribavirin, ombitasvir/paritaprevir/ritonavir/dasabuvir with or without ribavirin, elbasvir/grazoprevir (Zepatier), glecaprevir/pibrentasvir (Maviret), and sofosbuvir-based (SOF-based) regimens including sofosbuvir + ribavirin, sofosbuvir/ledipasvir (Harvoni), and sofosbuvir/velpatasvir (Epclusa). The present retrospective cohort study included patients with HCV who received DAA treatment at the Chiayi or Yunlin branches of Chang Gung Memorial Hospital between 1 January 2017 and 31 October 2020. This study was approved by the Research Ethics Committee of Chang Gung Memorial Hospital and was conducted per the principles of the Declaration of Helsinki and the International Conference on Harmonization for Good Clinical Practice guidelines. Patients with confirmed sustained virologic response (SVR) and available body weight data at the start of DAA therapy and at 2 years after achieving SVR were included in this study. An SVR was defined as having an HCV RNA level below the lower limit of quantification at least 12 weeks after the end of DAA therapy. The study design is shown in Figure 1. Patients who discontinued DAA therapy, had no SVR, died during the follow-up period, or had no available body weight data were excluded from this study.

### 2.2. Assessment of Clinical Parameters

The following data were extracted from the electronic medical records of eligible patients: (1) demographic characteristics; (2) body mass index (BMI) at baseline, 12 weeks after DAA completion (SVR12), 1 year after DAA completion, and endpoint; (3) results from laboratory examinations at baseline conducted to determine liver and renal function, fasting plasma glucose levels, glycated hemoglobin (HbA1c) levels, hemogram results, the prothrombin time-international normalized ratio (PT-INR), and alpha-fetoprotein (AFP) levels; (4) fibrosis (FIB)-4 scores at SVR12; and (5) DAA treatment details, including HCV viral profile and DAA treatment regimen. The degree of liver fibrosis was assessed indirectly using FIB-4 scores by applying the following formula: FIB-4 score = [age (years) × aspartate transaminase (AST) (U/L)]/[platelet count (10^9^/L) × alanine aminotransferase (ALT) (U/L)^1/2^]. A value of <1.45 was considered to indicate the absence of fibrosis or the presence of mild fibrosis, and a value of >3.25 was considered to indicate the presence of advanced fibrosis or cirrhosis. Post-treatment reductions in FIB-4 scores were calculated as the FIB-4 score at baseline minus the FIB-4 score at SVR12. The estimated glomerular filtration rate (eGFR) was calculated using the modification of diet in the renal disease equation.

Body weight change was defined as the difference between BMI at the baseline and endpoint. For baseline data, measurements taken within 90 days before the commencement of DAA therapy were considered, and the measurement taken closest to the start of therapy was used. For endpoint data, measurements taken within 90 days after the end of the follow-up were considered, and the measurement taken closest to the end of the follow-up was used. In cases in which data were not available, two measurements taken within 180 days after the end of the follow-up were interpolated. A study revealed that aging is associated with an expected weight gain of 1 to 2 lbs per year (approximately 0.45 to 0.90 kg per year), which is equivalent to an increase of approximately 1% of total body weight per year with aging [13]. There are some studies that discuss weight and adverse prognosis, including insulin resistance, metabolic syndrome, and related issues, and mention that when the weight increases by more than 5%, the risk begins to increase significantly. One study found that when comparing weight in middle age and age 20, every 5% increase in body weight was associated with a 20% greater risk of developing insulin resistance syndrome [14]. Another study found that when healthy adults gained more than 5% of their weight over a 5-year study period, the amount of weight gain was positively associated with the risk of metabolic syndrome [15]. With consideration of these, we established a 5% increase in BMI as the cutoff for defining a large weight gain during this 2-year follow-up study, which was also beyond the degree of weight gain expected with aging. BMI, calculated as a person’s weight in kilograms divided by the square of their height in meters, is widely employed to assess the severity of obesity. Standard cutoff values were used to categorize individuals as having underweight (<18.5 kg/m^2^), normal weight (18.5–24.9 kg/m^2^), overweight (25–29.9 kg/m^2^), obesity class I (30–34.9 kg/m^2^), and obesity class II/III (≥35 kg/m^2^).

### 2.3. Statistical Analysis

All statistical analyses were conducted using SPSS (version 22, IBM, Chicago, IL, USA). Continuous data were expressed as means (standard deviations), and categorical data were expressed as numbers (percentages). The chi-square test was employed to analyze categorical variables, and the Student’s *t*-test was used to analyze normally distributed continuous data. The paired *t*-test was used to compare baseline and endpoint BMIs. Univariate and stepwise multivariate logistic regression analyses were performed to evaluate factors related to BMI changes at the endpoint. The results are presented as odds ratios (ORs) with 95% confidence intervals (CIs). All statistical tests were two-tailed, and *p* < 0.05 was considered statistically significant.

## 3. Results

### 3.1. Baseline Characteristics of the Study Population

A total of 735 patients were enrolled in this study (Figure 2). The baseline characteristics of the study population are presented in Table 1. The mean age was 66.1 years, and 45.7% of the participants were men. The mean BMI was 25.6 kg/m^2^. The most prevalent weight category was normal (48.7%). The mean HCV-RNA titer was 10.9 × 10^6^ IU/mL and 493 (67.1%) patients were infected with HCV genotype 1b.

Comparing baseline data between two subgroups, patients with a large BMI increase (an increase of ≥5% from baseline) had a lower body weight (62.3 kg vs. 65.4 kg, *p* = 0.05), had a lower BMI (24.5 kg/m^2^ vs. 25.9 kg/m^2^, *p* < 0.001), had a higher ratio of normal BMI (62.0% vs. 44.9%, *p* < 0.001), lower ratio of overweight (27.6% vs. 38.1%, *p* = 0.014) or obesity class I (4.9% vs. 12.6%, *p* = 0.004), had a higher AST level (66.0 U/L vs. 55.7 U/L, *p* = 0.013), had a lower serum albumin level (4.20 g/dL vs. 4.30 g/dL, *p* = 0.030), had a lower platelet count (163.8 × 10^3^/μL vs. 182 × 10^3^/μL, *p* = 0.003), had a higher FIB-4 score (3.68 vs. 3.03, *p* = 0.006), had a higher FIB-4 score reduction (1.1 vs. 0.6, *p* = 0.002), had lower log-transformed HCV titers (5.7 vs. 6.0, *p* < 0.001), had a higher ratio of mixed genotype (5.5% vs. 2.3%, *p* = 0.039), and had a lower hemoglobin level (13.5 g/dL vs. 13.9 g/dL, *p* = 0.034) than the patients with no or a mild BMI increase (an increase of <5% from baseline).

### 3.2. BMI Change

The changes in BMI from baseline to the endpoint are presented in Figure 3. The mean BMI of the 735 patients was 25.56 ± 4.07 kg/m^2^ at baseline and 25.77 ± 4.29 kg/m^2^ at the endpoint (*p* = 0.005). For the patients who experienced a large BMI increase, the BMI curve continued to increase slowly throughout the tracking period. 

The distribution of the BMI change rate is presented in Figure 4. The average BMI change rate was 1.0% ± 8.3%. Specifically, the average BMI change rate was −0.5% ± 1.3% in the patients with no or a mild BMI increase and 2.78% ± 1.87% in the patients with a large BMI increase. For the top 10% of patients with the greatest weight gain, the average rate of BMI increase was 17.5% ± 9.9%.

The changes in weight categories at each time point (baseline, SVR12, 1 year after SVR, and 2 years after SVR [endpoint]) are presented in Figure 5. During the 2-year follow-up period, the proportion of patients with overweight and obesity increased from 49.4% to 54.3%.

The changes in weight category from baseline to the endpoint after stratification by weight category at baseline are presented in Figure 6. Of the 735 patients, 549 (74.6%) experienced no change in the weight category, whereas 111 (15.1%) were in a higher weight category at the endpoint. Among the patients with a normal BMI at baseline, at the endpoint, 19.6% had developed overweight, whereas 1% had developed obesity. Conversely, for the patients with overweight and obesity, more patients had an improved BMI than had a worse BMI. At the end of the follow-up period, 14.1% of the patients with overweight at baseline had a normal BMI. Among those with class I obesity at baseline, 1.3% had a normal BMI at the endpoint, and 25% were overweight.

### 3.3. Multivariate Regression Analysis for Notable BMI Increase

The predictors of a notable BMI increase are listed in Table 2. Univariate analyses revealed a lower body weight (*p* = 0.005), lower BMI (*p* < 0.001), higher AST level (*p* = 0.014), lower albumin level (*p* = 0.017), lower platelet count (*p* = 0.002), higher FIB-4 score reduction (*p* = 0.001), lower HCV titer (*p* < 0.001), and lower hemoglobin level (*p* = 0.035) at baseline to be associated with a considerable BMI increase at the endpoint. Additionally, individuals with overweight or obesity class I at baseline had a lower likelihood of experiencing a notable BMI increase than those with a normal BMI (OR: 0.525 and 0.283, respectively, *p* = 0.001). Multivariate analyses revealed that a higher FIB-4 score reduction (OR: 1.168; 95% CI: 1.047–1.304, *p* = 0.006) was associated with an increased risk of having a notably higher BMI at the endpoint. Conversely, older age (OR: 0.979; 95% CI: 0.963–0.996, *p* = 0.013) and a higher baseline BMI (OR: 0.907; 95% CI: 0.863–0.954, *p* < 0.001) were associated with a lower risk of having a notably higher BMI at the endpoint. Although the HCV titer was discovered to be significantly associated with a notable increase in BMI in multivariable regression, its OR was very close to 1.000, indicating that a unit increase in the predictor variable is likely to have a minimal effect on the outcome.

## 4. Discussion

In our study, patients with HCV who received DAA therapy and achieved SVR were noted to experience a significant increase in BMI after 2 years of follow-up. The average BMI change rate in the overall study population from baseline to the endpoint was 1.0% ± 8.3%, which is consistent with the rates reported in previous studies [9,10,11,12]. Research indicates that the weight changes that occur after DAA treatment are significantly different from those that occur after interferon treatment. Patients often experience varying degrees of weight loss during interferon treatment and return to their pretreatment weight 6 months after stopping interferon treatment [16]. To the best of our knowledge, no study has investigated the long-term weight changes that occur after interferon treatment.

The mechanisms underlying weight gain during DAA therapy remain under investigation. However, possible mechanisms include improvements in chronic inflammation, improvements in hepatic anabolic function, and improvements in quality of life after treatment [17,18,19]. The weight gain that occurs in patients undergoing DAA therapy may be similar to that of those undergoing combination antiretroviral therapy (cART) for HIV infection; patients undergoing cART often gain weight after treatment due to the reversal of the catabolic effects of HIV infection. After successful treatment, the weight distribution of individuals with HIV infection is similar to those without HIV infection [20,21].

In this study, a younger age, lower baseline BMI, and higher FIB-4 score reduction were predictive of a greater risk of having an increased BMI at the endpoint. The finding that younger patients are more susceptible to weight gain after HCV eradication is consistent with that of another study [10]. HCV infection adversely affects the body’s energy balance, resulting in a decrease in somatic and visceral protein amounts and an increase in body fat percentage [22,23]. Additionally, a study has reported a significant increase in skeletal muscle mass index after DAA treatment [24]. Given that the outcome measure in our study was BMI changes, we were unable to determine which part of the body composition was responsible for weight gain. However, if weight gain is assumed to be caused by an increase in skeletal muscle mass, then such a gain in mass could explain why younger adults gain more weight after treatment than older adults do. However, further research is required to validate this assumption.

Our finding that patients with lower baseline BMI were more likely to experience a notable increase in BMI after HCV eradication, possibly because such patients have lower anabolic status at baseline relative to other patients. When liver damage and infection burden are reduced or eliminated, patients with lower baseline BMI have more room to gain weight than other patients. However, our results were inconsistent with those of other studies that reported a higher BMI being associated with greater weight gain after therapy [10,12]. Different population ethnicities, cultural influences on health-promoting behaviors, and prevalences of chronic diseases may have contributed to these discrepancies. Our study revealed that overweight and obesity class I were associated with reduced odds of experiencing a notable increase in BMI after therapy. A study demonstrated that people with a BMI ≤ 30 are at risk of weight gain after HCV treatment, whereas people with a BMI > 30 do not gain weight [11]. This study assumed that people who are overweight or obese are physically inactive and, therefore, less likely to gain skeletal muscle mass and body weight. However, this assumption does not account for our observation that some individuals with overweight or obesity lost weight or even had a normal BMI at the endpoint. Our lack of data on body composition, exercise habits, diet, complete chronic medical history, and socioeconomic status limits our ability to explain this finding, and the underlying mechanism must be investigated in more extensive and detailed research.

Our finding that patients with higher reductions in FIB-4 scores are more likely to gain weight after HCV eradication has not been discussed in previous studies. In general, an eventual FIB-4 score reduction after HCV eradication can be considered to indicate fibrosis regression. However, a large reduction in FIB-4 score over a short period of time is generally caused by a reduction in transaminases, which is a consequence of inflammation remission rather than fibrosis resolution [25]. In such cases, after successful eradication, inflammation is relieved, and patients return to a relatively healthy state, which may lead to weight gain. The phenomenon that changes in inflammatory markers predicts disease prognosis is also seen in HIV-infected patients. For example, higher pretreatment C-reactive protein (CRP) levels indicate more severe disease states, and a greater decrease in CRP levels after HIV treatment indicates better disease control [26]. A study found that in underweight participants, an increase of 1 unit in body mass index was associated with a decrease of 9.32 mg/L in CRP levels [27].

During the observation period of this study, DAAs were not used. In other words, after the initial DAAs were completely metabolized, the DAAs themselves had almost no interference with weight changes. In the regression analysis of risk factors associated with weight gain, differences in DAA regimens also did not affect the results. Interestingly, we noted that the group with a large increase in BMI showed stable weight gain even during DAA treatment. In addition to the effect of DAA treatment, other confounding factors that may contribute to weight changes should also be considered, such as exercise habits, diet, or current medications. However, as this was a retrospective study, data on these confounding factors are incomplete. Additionally, we did not measure body weight regularly during treatment (e.g., once a week), so we do not know whether weight continued to increase throughout treatment, only started to increase later in treatment, or some other trajectory. The specific changes and possible causes need to be investigated through broader and more detailed studies in the future.

In the long term, we should evaluate whether people with increased BMI are at increased risk of diabetes, cardiovascular disease, or other metabolic diseases. If the risk of such diseases is unaffected or even reduced, it means that the patient is recovering from an unhealthy state. However, if the risk of such diseases does increase, it means that patients with identified predictors, such as younger age, lower baseline BMI, and higher FIB-4 score reduction, must pay greater attention to their weight after HCV eradication. Such patients should take a more proactive approach to avoid obesity, which may negate the benefits of HCV eradication.

This study has some limitations. First, this was a retrospective study, and the availability of weight data was limited by the irregularity of the post-treatment follow-up visits by the majority of the target patients, which then limited the number of patients that could be included. In addition, covariates that influence body weight, including exercise habits, diet, personal income, and education level, were not recorded and could not be assessed. Second, the observation period of our study was only two years, and longer-term monitoring of weight change may show a different trajectory. In addition, assessing the long-term effects of weight gain on hepatic and metabolic diseases, such as metabolic dysfunction-associated fatty liver disease, cirrhosis, HCC, diabetes, and cardiovascular disease, is crucial. Additionally, specific targets requiring timely intervention must be identified. Third, comparative studies should be conducted with matched cohorts of untreated hepatitis C patients. However, when DAA therapy is so effective, delaying HCV treatment solely for research purposes is not justified. In addition, there is no historical control group available for comparison. Fourth, BMI is an imperfect measure of obesity because it does not distinguish body fat from lean body mass. Previous studies have associated body fat with unfavorable health outcomes, whereas fat-free mass was associated with improved health [28]. Studies based on body composition rather than BMI may be appropriate to assess the complexities of body weight changes after HCV treatment.

## 5. Conclusions

This study revealed that treatment with DAA resulted in SVR and a significant increase in BMI after 2 years of follow-up in patients with HCV infection. A younger age, lower baseline BMI, and high FIB-4 score reduction after HCV treatment were associated with a more notable increase in BMI. Differences in DAA regimens did not affect outcomes. Future studies are required to elucidate the long-term effects and metabolic outcomes associated with such a change, and the exact mechanisms underlying the change must be investigated.

## Figures and Tables

**Figure 1 diagnostics-14-00213-f001:**
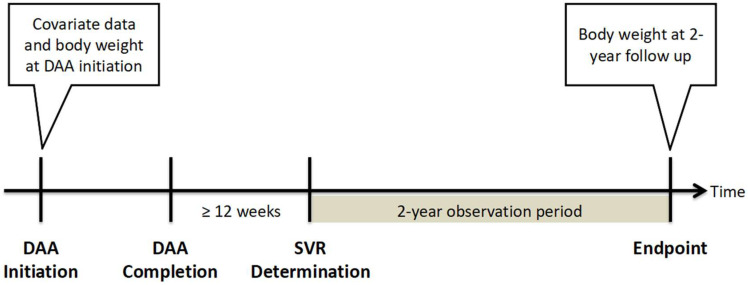
Study design. Abbreviation: DAA, direct-acting antiviral agent; SVR, sustained virologic response.

**Figure 2 diagnostics-14-00213-f002:**
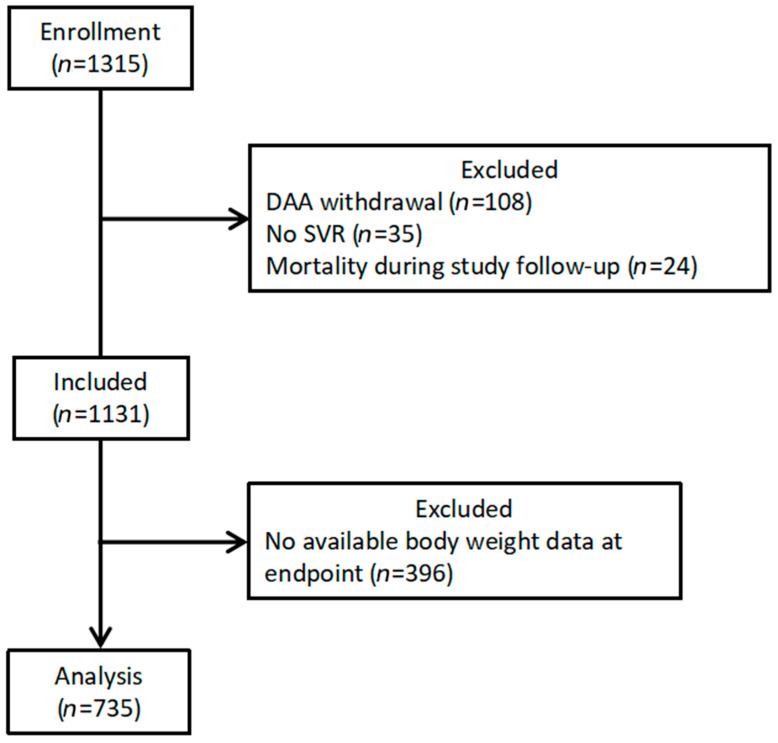
Patient selection flow diagram. Abbreviation: DAA, direct-acting antiviral agent; SVR, sustained virologic response.

**Figure 3 diagnostics-14-00213-f003:**
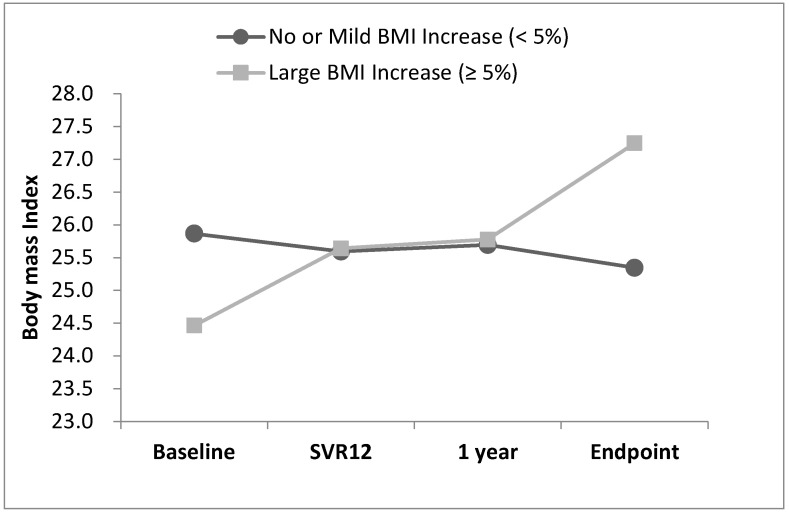
BMI trend from baseline to endpoint.

**Figure 4 diagnostics-14-00213-f004:**
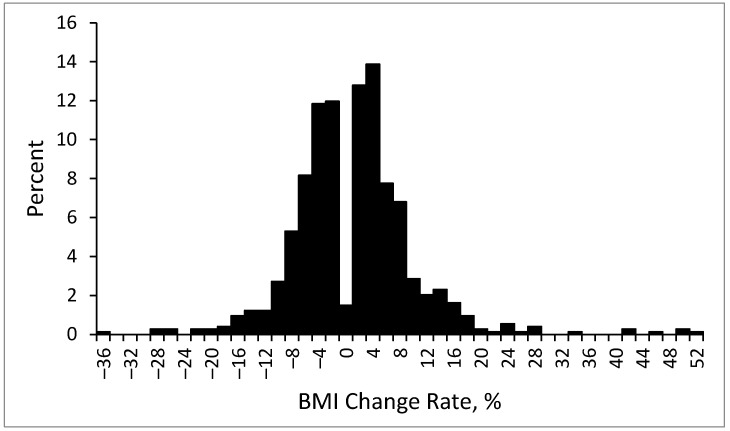
Distribution of BMI change rate at endpoint.

**Figure 5 diagnostics-14-00213-f005:**
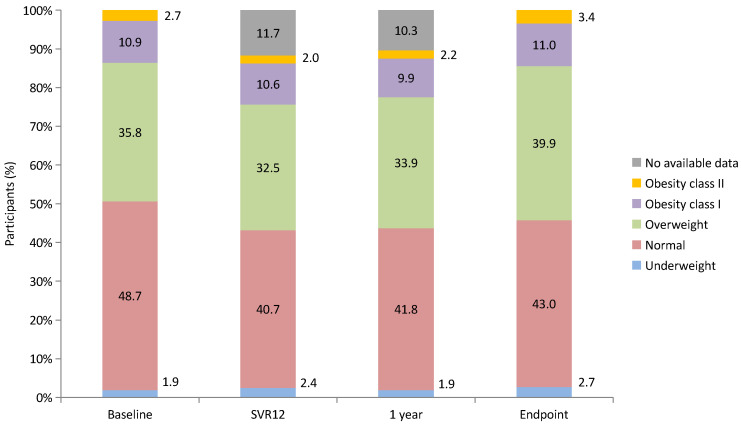
Changes in weight categories from baseline to endpoint.

**Figure 6 diagnostics-14-00213-f006:**
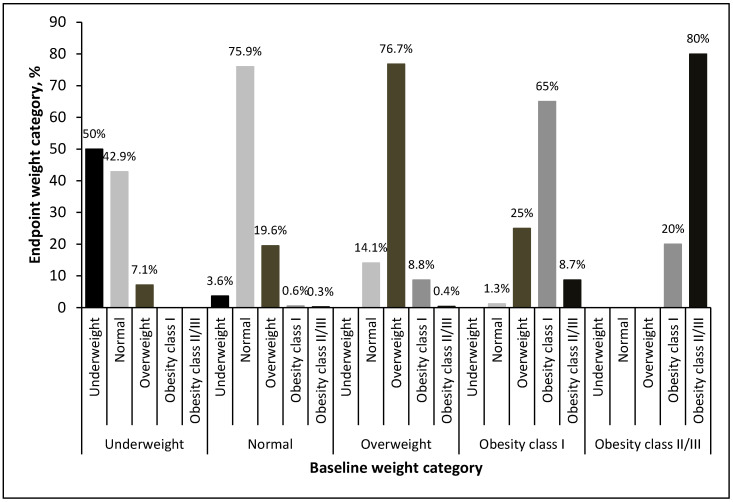
Weight category change at endpoint by weight category at baseline.

**Table 1 diagnostics-14-00213-t001:** Baseline characteristics.

	Total Patients (*n* = 735)	No or Mild BMI Increase (<5%)	*n*	Large BMI Increase (≥5%)	*n*	*p* Value
Age, year	66.1 (11.8)	66.3 (11.5)	572	65.6 (12.7)	163	0.525
Male gender, *n* (%)	336 (45.7%)	258 (45.1%)	572	78 (47.9%)	163	0.534
Body weight, kg	64.7 (12.4)	65.4 (12.1)	572	62.3 (13.0)	163	0.005
BMI, kg/m^2^	25.6 (4.1)	25.9 (4.0)	572	24.5 (4.0)	163	<0.001
Weight category, *n* (%)			572		163	
Underweight: <18.5 kg/m^2^	14 (1.9%)	9 (1.6%)		5 (3.1%)		0.208
Normal: 18.5–24.9 kg/m^2^	358 (48.7%)	257 (44.9%)		101 (62.0%)		<0.001
Overweight: 25.0–29.9 kg/m^2^	263 (35.8%)	218 (38.1%)		45 (27.6%)		0.014
Obesity class I: 30.0–34.9 kg/m^2^	80 (10.9%)	72 (12.6%)		8 (4.9%)		0.004
Obesity class II and III: >35.0 kg/m^2^	20 (2.7%)	16 (2.8%)		4 (2.5%)		1.000
Current smoking, *n* (%)	138 (18.8%)	105 (18.4%)	572	33 (20.2%)	163	0.586
Diabetes mellitus, *n* (%)	199 (27.1%)	148 (25.9%)	572	51 (31.3%)	163	0.170
Hypertension, *n* (%)	367 (49.9%)	284 (49.7%)	572	83 (50.9%)	163	0.775
Dyslipidemia, *n* (%)	153 (20.8%)	114 (19.9%)	572	39 (23.9%)	163	0.268
HBsAg positive, *n* (%)	98 (13.3%)	79 (13.8%)	572	19 (11.7%)	163	0.475
HCC, *n* (%)	50 (6.8%)	41 (7.2%)	572	9 (5.5%)	163	0.597
Prior IFN, *n* (%)	155 (21.1%)	121 (21.2%)	254	34 (20.9%)	65	0.501
AST, U/L	58.0 (46.5)	55.7 (45.7)	572	66.0 (48.6)	163	0.013
ALT, U/L	72.1 (68.7)	70.5 (71.1)	572	77.8 (59.3)	163	0.228
Albumin, g/dL	4.3 (0.4)	4.3 (0.4)	572	4.2 (0.4)	163	0.030
Albumin ≤ 3.5 g/dL, *n* (%)	23 (3.1%)	12 (2.1%)	572	11 (6.7%)	163	0.003
Total bilirubin, mg/dL	0.8 (0.4)	0.7 (0.4)	572	0.8 (0.6)	163	0.173
Platelet, 10^3^/μL	178.0 (65.6)	182.0 (67.0)	572	163.8 (58.5)	163	0.002
PT INR	1.0 (0.1)	1.0 (0.1)	572	1.0 (0.2)	163	0.040
FIB-4 score	3.17 (2.45)	3.03 (2.36)	572	3.68 (2.74)	163	0.006
FIB-4 score, *n* (%)			572		163	
<1.45	170 (23.1%)	142 (24.8%)		28 (17.2%)		0.041
1.45–3.25	306 (41.6%)	246 (43.0%)		60 (36.8%)		0.157
>3.25	259 (35.2%)	184 (32.2%)		75 (46.0%)		0.001
FIB-4 score reduction *	0.8 (1.7)	0.6 (1.6)	566	1.1 (1.8)	161	0.002
AFP, ng/mL	8.0 (28.3)	7.4 (26.8)	572	10.3 (33.0)	163	0.247
HCV-RNA, 10^6^ IU/mL	10.9 (208.4)	13.5 (236.2)	572	2.0 (2.7)	163	0.538
HCV-RNA, log-transformed	6.0 (1.0)	6.0 (0.9)	572	5.7 (1.0)	163	<0.001
HCV genotype, *n* (%)			572		163	
1a	9 (1.2%)	6 (1.0%)		3 (1.8%)		0.423
1b	493 (67.1%)	383 (67.0%)		110 (67.5%)		0.900
2	201 (27.3%)	162 (28.3%)		39 (23.9%)		0.267
3	2 (0.3%)	1 (0.2%)		1 (0.6%)		0.395
6	8 (1.1%)	7 (1.2%)		1 (0.6%)		0.692
mixed	22 (3.0%)	13 (2.3%)		9 (5.5%)		0.039
DAAs, *n* (%)			572		163	
DCV/ASV	41 (5.6%)	29 (5.1%)		12 (7.4%)		0.261
ProD	98 (13.3%)	76 (13.3%)		22 (13.5%)		0.944
Zepatier	271 (36.9%)	220 (38.5%)		51 (31.3%)		0.094
Maviret	105 (14.3%)	81 (14.2%)		24 (14.7%)		0.856
SOF-based	220 (29.9%)	166 (29.0%)		54 (33.1%)		0.312
Ribavirin use, *n* (%)	84 (11.4%)	61 (10.7%)	572	23 (14.1%)	163	0.222
Hemoglobin, g/dL	13.8 (1.7)	13.9 (1.7)	572	13.6 (1.8)	163	0.034
Fasting plasma glucose, mg/dL	115.4 (39.66)	113.9 (39.4)	318	120.7 (40.4)	90	0.149
Creatinine, mg/dL	0.91 (0.36)	0.91 (0.37)	572	0.89 (0.29)	163	0.493
eGFR, mL/min/1.73 m^2^	84.35 (23.99)	83.88 (23.94)	572	85.99 (24.18)	163	0.323

Data are expressed as means (standard deviations) or numbers (percentages). Abbreviations: ALT, alanine aminotransferase; AFP, alpha-fetoprotein; AST, aspartate transaminase; BMI, body mass index; DAA, direct-acting antiviral agent; DCV/ASV, daclatasvir/asunaprevir; eGFR, estimated glomerular filtration rate; FIB-4, fibrosis index based on four factors; HCC, hepatocellular carcinoma; IFN, interferon; ProD, paritaprevir/ritonavir/ombitasvir/dasabuvir; SOF, sofosbuvir. * FIB-4 score reduction = FIB-4 score at baseline − FIB-4 score at SVR determination.

**Table 2 diagnostics-14-00213-t002:** Logistic regression analysis for notable BMI increase.

	Univariate		Multivariate	
Variables	OR (95%CI)	*p* Value	OR (95%CI)	*p* Value
Age (year)	0.995 (0.981–1.010)	0.525	0.979 (0.963–0.996)	0.013
Gender (Male vs. Female)	1.117 (0.788–1.583)	0.535	1.120 (0.743–1.689)	0.587
Body weight (kg)	0.979 (0.964–0.993)	0.005		
BMI (kg/m^2^)	0.910 (0.868–0.955)	<0.001	0.907 (0.863–0.954)	<0.001
Weight category				
Underweight: <18.5 kg/m^2^	1.414 (0.463–4.320)	0.544		
Normal: 18.5–24.9 kg/m^2^	Reference			
Overweight: 25.0–29.9 kg/m^2^	0.525 (0.354–0.780)	0.001		
Obesity class I: 30.0–34.9 kg/m^2^	0.283 (0.131–0.608)	0.001		
Obesity class II and III: ≧35.0 kg/m^2^	0.636 (0.208–1.949)	0.428		
Current smoking (Yes vs. No)	1.129 (0.729–1.747)	0.586		
Diabetes Mellitus (Yes vs. No)	1.305 (0.892–1.908)	0.171	1.092 (0.693–1.720)	0.0704
Hypertension (Yes vs. No)	1.052 (0.743–1.490)	0.775	1.317 (0.867–2.002)	0.197
Dyslipidemia (Yes vs. No)	1.264 (0.835–1.912)	0.268	1.355 (0.841–2.184)	0.212
HBsAg positive (Yes vs. No)	0.823 (0.483–1.405)	0.476		
HCC (Yes vs. No)	0.757 (0.360–1.592)	0.463		
AST (U/L)	1.004 (1.001–1.008)	0.014		
ALT (U/L)	1.001 (0.999–1.004)	0.231		
Albumin (g/dL)	0.582 (0.373–0.907)	0.017	0.819 (0.491–1.367)	0.445
Total bilirubin (mg/dL)	1.280 (0.886–1.849)	0.189		
Platelet (10^3^/μL)	0.995 (0.993–0.998)	0.002	0.996 (0.993–0.999)	0.022
FIB-4 score (Baseline)	1.102 (1.033–1.176)	0.003		
<1.45	Reference			
1.45–3.25	1.237 (0.755–2.027)	0.399		
>3.25	2.067 (1.271–3.361)	0.003		
FIB-4 score (SVR12)	1.054 (0.966–1.149)	0.236		
<1.45	Reference			
1.45–3.25	0.847 (0.561–1.281)	0.432		
>3.25	1.390 (0.847–2.282)	0.193		
FIB-4 score reduction ^a^	1.189 (1.072–1.319)	0.001	1.168 (1.047–1.304)	0.006
AFP (ng/mL)	1.003 (0.998–1.008)	0.283		
HCV-RNA (10^6^ IU/mL)	0.892 (0.837–0.950)	<0.001	1.000 (1.000–1.000)	0.006
HCV genotype				
1a	NA ^b^			
1b	Reference			
2	0.838 (0.557–1.262)	0.398		
3	NA ^b^			
6	NA ^b^			
mixed	NA ^b^			
DAAs				
DCV/ASV	1.272 (0.607–2.665)	0.524		
ProD	0.890 (0.506–1.566)	0.686		
Zepatier	0.713 (0.462–1.098)	0.125		
GP	0.911(0.526–1.578)	0.739		
SOF-based	Reference			
SOF-based (Yes vs. No)	1.212 (0.835–1.759)	0.313		
Ribavirin use (Yes vs. No)	1.376 (0.823–2.303)	0.224		
Hemoglobin (g/dL)	0.896 (0.809–0.992)	0.035	0.908 (0.801–1.030)	0.133
Fasting plasma glucose (mg/dL)	1.004 (0.999–1.009)	0.154		
HbA1c (%)	1.036 (0.858–1.251)	0.713		
Creatinine (mg/dL)	0.831 (0.490–1.411)	0.494		
eGFR (mL/min/1.73 m^2^)	1.004 (0.996–1.011)	0.323		

NA, not applicable. ^a^ FIB-4 score reduction = FIB-4 score at baseline − FIB-4 score at SVR determination ^b^ Too few cases.

## Data Availability

The data presented in this study are available on request from the corresponding author.

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
