# Peer review of "Weight Gain and Increased Body Mass Index in Patients with Hepatitis C after Eradication Using Direct-Acting Antiviral Therapy in Taiwan"

_diagnostics, 2024, doi:10.3390/diagnostics14020213_

Round 1

Reviewer 1 Report

Comments and Suggestions for Authors

This was a retrospective study of 735 patients with chronic hepatitis C infection seen at 2 hospitals in Taiwan who achieved SVR after receiving DAA treatment. The primary outcome was change in BMI 2 years after SVR. The main finding was that younger age, lower baseline BMI, and higher FIB-4 score reduction were associated with increased BMI (>5%) at endpoint. The clinical implication would be close monitoring of diet, exercise, weight, and metabolic parameters such as hemoglobin A1c and lipids in normal weight patients after DAA therapy. A major limitation of the study is that about one-third of patients did not have a 2 year weight available. 

This is an interesting study. Given Figure 3 showing steady weight gain in the large BMI change group even during DAA treatment, I am not sure if the effects the authors are seeing are related to HCV eradication. This paper presents a lot of different analyses and ways of looking at a relatively small number of key findings. Simplifying the message would help it have more impact. 

Major comments

1.     How did the authors decide that 5% increase in BMI was “large”? Why was this the primary outcome as opposed to 5-10% body weight change or change in BMI category? 

2.     What do the authors hypothesize is the reason that patients with large BMI increase had a greater reduction in FIB-4 score over the study?

3.     Figure 3: interesting that the group with large BMI change shows steady weight gain even during DAA treatment. Any hypothesis about this? Are these patients who are just gaining weight due to lifestyle factors, other medical problems, or medications, and it has nothing to do with DAA treatment?

4.     Line 162: “for the patients with no or a mild BMI increase, the BMI curve remained nearly unchanged.” Don’t think this needs to be said, already implied based on the fact that BMI did not change significantly during the study.

5.     Line 168: what does “mean BMI change rate” mean? Over what time frame is that percentage? (In other words, mean BMI change rate is listed as 1%, is that yearly or over the whole study?) The Discussion mentions over the whole study, so if this is the case it shoud be clear in the Results. Overall, I am not sure what BMI change rate adds to the message of the paper. 

6.     Table 2: can probably get rid of this table, does not really add to the overall message and 

7.     Figure 5: the SVR12 and 1 year bars do not add up to 100%--it was stated in the manuscript that not all patients had weights at these time points (649 and 659 patients respectively), but the figure is confusing because those 2 bars go all the way up to 100% on the y-axis scale. The figure is overall hard to understand because the 43% bar for Endpoint is therefore smaller than the 40.7% and 41.8% bars in the middle time points. Would recommend adjusting heights of bars so that the height accurately represents the percentage. Could also list the N for each time point on the x-axis to be clear what the percentage represents. 

8.     Figure 6: needs better labels. Are the labels on the x-axis the baseline categories, and then bars going up are the percent who ended up in each category? 

Reviewer 2 Report

Comments and Suggestions for Authors

Dear Authors,

The current study reveals weight gain and increased body mass index in patients with hepatitis C after eradication using direct-acting antiviral therapy in Taiwan. Good efforts had been done in that study. However, there are some comments:

First: The authors are looking for factors accompanying weight gain after DAA treatment. Accordingly, regarding the study design, prospective study is appropriate than retrospective study in that situation.

Second: the author did not mention the control group.

Third: the patients’ diet and psychological condition had not been evaluated in detail.

Fourth: the author should take into consideration the effect of the regimen that had been used in the present work.

Fifth: Further studies should be performed to compare different regimens that used in different countries.

Sixth: The current work didn’t demonstrate the stability of these changes in BMI and their effect on hepatic steatosis and fibrosis and to evaluate the effect of this increase in BMI on the outcomes of HCV treatment.

Seventh: The Author didn’t define all outcomes for which data were sought.

Eighth: There are some linguistic and grammatical errors that must be rephrased and written in a correct style.

Good Luck

Comments on the Quality of English Language

Moderate editing of English language required

Reviewer 3 Report

Comments and Suggestions for Authors

Our esteemed authors

As is known

Hepatitis C was a serious health problem for the whole world. In recent years, with the efforts of excellent physicians like you, many negative conditions such as viral hepatitis C and its related cirrhosis, chronic liver disease and even hepatocellular cancer could be prevented, and HCV eradication could be achieved.

Wonderful drugs such as DAAs, which provide relief from such an important health problem, are said to cause an increase in body mass index and weight gain in some patient groups, in addition to their countless benefits.

In this very valuable study you conducted on a large group of patients treated with DAAs for 3 years, it was found that it caused weight gain and body mass increase in some patients, while it was affected by the initial patient body mass index, age, and other parameters. It has been shown that there is no effect.

By publishing this beautiful study in a good journal, it will make significant contributions to the education and patient approach of both medical students, residents and specialists.

I thank you very much for this beautiful work and wish you continued success.

Round 2

Reviewer 2 Report

Comments and Suggestions for Authors

Dear Author,

Good Job. I hope that you will amend your manuscript, correct all the comments mentioned by the reviewers, and work to add them to the research paper.

Best of Luck

Comments on the Quality of English Language

 Minor editing of English language required